# Strategic and Chemical Advances in Antibody–Drug Conjugates

**DOI:** 10.3390/pharmaceutics17091164

**Published:** 2025-09-05

**Authors:** Ibrahim A. Alradwan, Meshal K. Alnefaie, Nojoud AL Fayez, Alhassan H. Aodah, Majed A. Majrashi, Meshael Alturki, Mohannad M. Fallatah, Fahad A. Almughem, Essam A. Tawfik, Abdullah A. Alshehri

**Affiliations:** 1Advanced Diagnostics and Therapeutics Institute, Health Sector, King Abdulaziz City for Science and Technology (KACST), Riyadh 11442, Saudi Arabia; ialradwan@kacst.gov.sa (I.A.A.); malnefaie@kacst.gov.sa (M.K.A.); nalfayez@kacst.gov.sa (N.A.F.); aaodah@kacst.gov.sa (A.H.A.); mfallatah@kacst.gov.sa (M.M.F.); falmughem@kacst.gov.sa (F.A.A.); 2Bioengineering Institute, Health Sector, King Abdulaziz City for Science and Technology (KACST), Riyadh 11442, Saudi Arabia; mmajrashi@kacst.gov.sa; 3Wellness and Preventative Medicine Institute, Health Sector, King Abdulaziz City for Science and Technology (KACST), Riyadh 11442, Saudi Arabia; mmsalturki@kacst.gov.sa

**Keywords:** antibody–drug conjugates, site-specific conjugation, lysine conjugation, cysteine conjugation, enzymatic tagging, glycan remodeling, non-canonical amino acids, drug-to-antibody ratio, linker chemistry, targeted cancer therapy

## Abstract

Antibody–drug conjugates (ADCs) are a rapidly advancing class of targeted cancer therapeutics that couple the antigen specificity of monoclonal antibodies (mAbs) with the potent cytotoxicity of small-molecule drugs. In their core design, a tumor-targeting antibody is covalently linked to a cytotoxic payload via a chemical linker, enabling the selective delivery of highly potent agents to malignant cells while sparing normal tissues, thereby improving the therapeutic index. Humanized and fully human immunoglobulin G1(IgG1) antibodies are the most common ADC backbones due to their stability in systemic circulation, robust Fcγ receptor engagement for immune effector functions, and reduced immunogenicity. Antibody selection requires balancing tumor specificity, internalization rate, and binding affinity to avoid barriers to tissue penetration, such as the binding-site barrier effect, while emerging designs exploit tumor-specific antigen variants or unique post-translational modifications to further enhance selectivity. Advances in antibody engineering, linker chemistry, and payload innovation have reinforced the clinical success of ADCs, with more than a dozen agents FDA approved for hematologic malignancies and solid tumors and over 200 in active clinical trials. This review critically examines established and emerging conjugation strategies, including lysine- and cysteine-based chemistries, enzymatic tagging, glycan remodeling, non-canonical amino acid incorporation, and affinity peptide-mediated methods, and discusses how conjugation site, drug-to-antibody ratio (DAR) control, and linker stability influence pharmacokinetics, efficacy, and safety. Innovations in site-specific conjugation have improved ADC homogeneity, stability, and clinical predictability, though challenges in large-scale manufacturing and regulatory harmonization remain. Furthermore, novel ADC architectures such as bispecific ADCs, conditionally active (probody) ADCs, immune-stimulating ADCs, protein-degrader ADCs, and dual-payload designs are being developed to address tumor heterogeneity, drug resistance, and off-target toxicity. By integrating mechanistic insights, preclinical and clinical data, and recent technological advances, this work highlights current progress and future directions for next-generation ADCs aimed at achieving superior efficacy, safety, and patient outcomes, especially in treating refractory cancers.

## 1. Introduction

The concept of targeted cancer therapy has undergone a remarkable evolution over the past century with the development of antibody–drug conjugates (ADCs). Targeted therapy has revolutionized cancer treatment by shifting the paradigm from non-specific cytotoxic agents to precision-guided interventions that exploit molecular vulnerabilities unique to the cancer cells [1,2]. Among the most significant advancements in this field is the emergence of ADCs, a novel class of therapeutics that couples high selectivity of monoclonal antibodies (mAbs) with chemotherapeutic agents [2]. This combination enables the delivery of ultra-potent cytotoxic payloads directly to cancer cells while minimizing off-target effects to improve the therapeutic index [3]. Since the approval of the first ADC, gemtuzumab ozogamicin, in 2000 for acute myeloid leukemia (AML), the field has grown rapidly [4]. As of 2025, ~19 ADCs have received global market approval for various hematologic and solid malignancies [5]. Additionally, more than 200 ADC candidates are currently being evaluated in Phase I-III clinical trials, highlighting the expanding clinical and commercial interest in this transformative therapeutic modality [5]. The global ADC market was valued at approximately USD 7.82 billion in 2022 [6]. Technological advancements in antibody humanization, linker stability, site-specific conjugation, and payload diversity have addressed many of the early limitations of ADCs, such as off-target toxicity, suboptimal drug-to-antibody ratio (DAR), and heterogeneous pharmacokinetics, to produce more effective and safer ADCs [6,7].

A typical ADC consists of three essential components: a mAb that selectively targets a tumor antigen; a potent cytotoxic payload; and a chemical linker that connects the two, which governs the pharmacokinetics and stability of the conjugate [8]. The antibody component is usually a fully human or humanized IgG, commonly of the IgG1 subclass, selected due to its favorable half-life and ability to engage immune effector functions such as antibody-dependent cellular cytotoxicity (ADCC) and phagocytosis, but IgG2, IgG4, nanobody, and bispecific antibody can also be used [8,9]. The target antigen must be highly expressed on cancer cells and minimally present on normal tissues to reduce on-target/off-tumor toxicity [9]. The cytotoxic payloads employed in current ADCs are significantly more potent than standard chemotherapeutics, with half-maximal inhibitory concentration (IC_50_) values in the low nanomolar or picomolar range [10,11].

Consistent with these observations, several U.S. Food and Drug Administration (FDA)-approved ADCs exhibit sub-nanomolar to picomolar cytotoxic potency, reflecting the highly selective and effective nature of their payloads (Table 1). For example, belantamab mafodotin (approved August 5, 2020) demonstrates an IC_50_ of approximately 3 ng/mL, while loncastuximab tesirine (approved April 23, 2021) achieves an IC_50_ as low as 4 pM [12]. Other notable examples include ado-trastuzumab emtansine (0.24 nM), polatuzumab vedotin (0.07 nM), and gemtuzumab ozogamicin (0.03 nM), each highlighting the extreme potency and therapeutic precision achievable through optimized payload–antibody combinations [13].

DM1 and MMAE are two of the most commonly used cytotoxic payloads in FDA-approved ADCs, both acting as microtubule inhibitors that induce mitotic arrest and apoptosis [14]. Despite their similar functional outcomes, they differ in origin, structure, and binding sites: DM1, a maytansine derivative, binds the maytansine site on β-tubulin, while MMAE, a synthetic analog of dolastatin 10, targets the vinca alkaloid site. These agents also differ in conjugation strategies; DM1 is typically linked via non-cleavable linkers (e.g., in T-DM1), whereas MMAE is delivered through cleavable dipeptide linkers (e.g., in brentuximab vedotin). These design choices influence intracellular release, stability, bystander killing, and therapeutic index, making the selection of payload–linker combinations a critical component of ADC optimization.

The linker has a critical role in ensuring payload release is selective within the tumor environment. Both cleavable and non-cleavable linkers can be used; cleavable linkers respond to conditions such as low pH or protease activity within lysosomes, while non-cleavable linkers rely on complete lysosomal degradation of the antibody [15]. Among the FDA-approved ADCs, targets include cluster of differentiation 33 (CD33), CD22, CD79b, Nectin 4, human epidermal growth factor receptor 2 (HER2), trophoblast cell surface antigen 2 (TROP2), and B-cell maturation antigen (BCMA), with expression patterns tailored to specific hematologic and solid tumor indications [16].

Mechanistically, ADCs leverage mAb to deliver potent cytotoxic agents directly to cancer cells (Figure 1). Once administered, the antibody component binds selectively to the tumor-associated antigens that are overexpressed on the surface of malignant cells, such as HER2 in breast cancer, CD33 in AML, or TROP2 in triple-negative breast cancer (TNBC) [17]. This antigen recognition is mediated by the antigen-binding fragment (Fab) region of the antibody, particularly through the complementarity-determining regions (CDRs) located within the variable domains of both heavy and light chains [18]. These CDRs interact specifically with unique epitopes on the target antigen, enabling high-affinity and selective binding [18]. Following antigen engagement, the ADC antigen complex is internalized via receptor-mediated endocytosis and trafficked to the lysosomal compartment [19]. There, cleavage of the linker, facilitated by enzymatic activity, acidic pH, or reducing conditions, triggers the intracellular release of the cytotoxic payload [19]. The released cytotoxic agent, whether a microtubule inhibitor or a topoisomerase I inhibitor, then induces apoptosis through mechanisms such as mitotic arrest or the formation of double-stranded DNA breaks [15]. This process enables localized tumor cell killing, reducing the systemic toxicities traditionally associated with chemotherapy, such as severe immunosuppression, gastrointestinal mucositis, and peripheral neuropathy [15]. Several ADCs exhibit a “bystander effect,” whereby membrane-permeable payloads (e.g., MMAE, SN-38) can diffuse into adjacent tumor cells that lack the target antigen, thereby enhancing efficacy in tumors with heterogeneous expression [15]. This review critically examines the current landscape and future directions of ADC development, emphasizing innovations in molecular design and conjugation chemistry that are redefining the role of ADCs in oncology and beyond.

## 2. Historical Development and ADC Overview

For nearly a century, cancer treatment has predominantly relied on systemic chemotherapy, employing cytotoxic agents that inhibit cell division and induce cell death. While these agents have improved survival rates, their lack of specificity often causes significant side effects by targeting not only cancer cells but also rapidly dividing healthy tissues [20]. Inspired by Paul Ehrlich’s “magic bullet” vision of selectively eradicating cancer cells while sparing healthy tissue, this vision became scientifically plausible in 1978, when Köhler and Milstein’s hybridoma technology enabled the large-scale production of antigen-specific monoclonal antibodies (mAbs), marking a significant milestone in the field [21,22]. Early proof-of-concept studies in the late 1970s and early 1980s investigated the chemical linkage of monoclonal antibodies (mAbs) to potent cytotoxins, such as ricin A-chain, saporin, or conventional chemotherapeutics. These first-generation constructs, while conceptually promising, were limited by the use of murine antibodies, unstable linkers, and heterogeneous DARs [20,23,24].

The first human clinical trial of an ADC was initiated in 1983 with BR96-doxorubicin, targeting the Lewis Y (Ley) antigen, which confirmed the feasibility of mAb-mediated targeted delivery but also highlighted the challenges of linker stability, tumor penetration, and off-target toxicity [25]. The field reached its first regulatory milestone in 2000 with the FDA approval of gemtuzumab ozogamicin (Mylotarg^®^), an anti-CD33 ADC for acute myeloid leukemia, which used a hydrazone linker to connect a humanized mAb to the DNA-damaging agent calicheamicin [22]. Although this marked a turning point, safety concerns, most notably hepatotoxicity and veno-occlusive disease, led to its withdrawal in 2010 [24]. This setback drove the development of second-generation ADCs with humanized antibodies, more stable linkers, and highly potent payloads, culminating in the approval of ado-trastuzumab emtansine (T-DM1) in 2013 for HER2-positive metastatic breast cancer [9,24]. The next breakthrough came in 2019 with trastuzumab deruxtecan (DS-8201), which achieved unprecedented efficacy, including in HER2-low breast cancer, by combining a cleavable tetrapeptide linker with a membrane-permeable topoisomerase I inhibitor payload [9]. As of 2025, the ADC landscape includes 19 FDA-approved products and over 200 clinical candidates, targeting a diverse range of antigens across hematologic and solid tumors. These advances reflect four decades of iterative progress in antibody engineering, linker chemistry, and payload optimization, transforming ADCs from experimental “magic bullets” into a cornerstone of precision oncology [9].

Therapeutic mAbs have progressed from murine to chimeric, humanized, and fully human forms, which reduces immunogenicity and improves efficacy. The murine mAbs are limited by high immune responses and short half-lives, while chimeric mAbs, combining human constant regions with murine variable regions, partially addressed these issues but still elicited immune reactions. The fully humanized mAbs further minimized murine sequences to complementarity-determining regions, and fully human mAbs now offer optimal tolerability and therapeutic performance [20].

Most ADCs in clinical use or development incorporate humanized or fully human mAbs, as these provide high target specificity, long serum half-lives, and minimal immunogenicity. Exceptions, such as brentuximab vedotin, utilize chimeric antibodies, although less common [20,25]. The IgG1 subclass is predominantly used in ADC development owing to its stability and ability to trigger immune responses. Other subclasses, such as IgG2, IgG3, and IgG4, are less favored due to inherent limitations [25]. Key factors influencing the selection of mAbs for ADCs include antigen specificity, binding affinity, immunogenicity, and pharmacokinetics. However, the relationship between mAb properties and ADC efficacy remains complex, with challenges arising from immunogenicity and target-binding dynamics, emphasizing the need for continued optimization of mAbs for ADC applications [20]. Binding of IgG-based ADCs to the neonatal Fc receptor (FcRn) at acidic endosomal pH (~6.0) prevents lysosomal degradation by mediating their recycling to the circulation, where dissociation occurs at physiological pH (~7.4), thereby prolonging half-life and systemic exposure [21]. The transition from murine mAbs to chimeric and humanized antibodies, ultimately yielding fully human IgGs, further enhanced therapeutic performance by extending circulation time via FcRn interaction and increasing antibody-dependent cellular cytotoxicity (ADCC) activity [21].

To produce fully human mAbs, modern techniques such as phage display and transgenic mouse technologies are employed. These approaches involved engineering mice to generate human antibodies instead of murine ones, thereby reducing immunogenicity and enhancing clinical efficacy [21]. The mAbs can be categorized into four types based on their origin and structure [25]. Murine antibodies (“-onab”) are entirely derived from mouse genes, including both light and heavy chains. While they were the first generation of mAbs, they suffer from high immunogenicity, short half-lives, and poor tumor penetration, as seen with Muromonab [25]. Chimeric antibodies (“-ximab”) are 35% mouse and 65% human, combining murine variable regions with human constant regions [25]. This design reduces immunogenicity and extends half-lives compared to murine antibodies. The progression of antibody engineering from chimeric to humanized antibodies (e.g., rituximab and alemtuzumab) and ultimately to fully human antibodies (e.g., adalimumab) demonstrates the shift toward increasing human sequence content to reduce immunogenicity and enhance therapeutic efficacy [21,25].

ADCs’ development hinges on the careful selection of an appropriate target antigen, as this choice is crucial to their therapeutic effectiveness. An ideal target antigen should exhibit tumor-specific expression, uniform distribution, high expression levels, efficient internalization, and minimal ectodomain shedding [25]. However, most antigens identified for ADCs are tumor-associated rather than strictly tumor-specific. Optimal antigens are typically expressed in tissues that are either resistant to the ADC payload or possess robust regenerative capabilities [23]. Crucially, the level of antigen expression directly influences ADC efficacy, as it determines the amount of cytotoxic payload delivered into cancer cells [9].

Natural products have played a pivotal role in advancing ADC technology, with auristatins and their derivatives (e.g., MMAE) standing out as key examples. Derived from marine cytotoxins, auristatins enhance the pharmacokinetics of ADCs and prolong their half-lives. Originally derived from marine cytotoxins, auristatins serve as potent tubulin inhibitors that disrupt microtubule dynamics, induce cell cycle arrest, and hence enhance ADC efficacy. The cytotoxic payloads employed in ADCs generally fall into three major classes [25]. Calicheamicins, enediyne antibiotics produced by Micromonospora echinospora, induce DNA double-strand breaks leading to apoptosis and are incorporated into FDA-approved ADCs such as Mylotarg^®^ and Besponsa^®^ [25]. Auristatins, sourced from marine organisms such as sea hares, function as tubulin inhibitors, with MMAE being the most clinically prevalent, featured in ADCs including Adcetris^®^ and Padcev^®^ [25]. Maytansinoids, derived from the plant Maytenus serrata, inhibit tubulin polymerization and cause mitotic arrest, exemplified by the use of DM1 in Kadcyla^®^ [25].

In solid tumors, ADC effectiveness typically increases with antigen density, but those with cleavable linkers and membrane-permeable payloads, or non-internalizing ADCs activated externally, can remain effective even when antigen internalization is limited [9]. Emerging ADC payloads present novel therapeutic opportunities by exploiting distinct mechanisms of cytotoxicity. Amanitins, naturally occurring bicyclic peptides, inhibit RNA polymerase II, thus blocking transcription and effectively targeting both proliferating and quiescent tumor cells. Heidelberg Pharma’s HDP-101 is a leading candidate in this class [25]. Duocarmycins are potent DNA-alkylating agents that bind preferentially to adenine residues in the minor groove, inducing irreversible DNA damage and apoptosis; trastuzumab duocarmazine, developed by Byondis, is currently undergoing Phase III clinical trials [25]. Pyrrolobenzodiazepines (PBDs) are highly cytotoxic DNA-interactive agents that form covalent bonds with guanine bases, generating DNA cross-links that trigger apoptosis. Several PBD dimer-containing ADCs are in advanced development by ADC Therapeutics [24].

By integrating these highly potent cytotoxic agents with advanced targeting mechanisms, ongoing research is addressing current limitations to position ADCs as a cornerstone of cancer therapy [25]. Minimal antigen shedding is another critical factor in ADC development. Excessive shedding can lead to circulating antigens binding to ADCs, reducing the availability of the drug for tumor targeting and necessitating higher doses [9]. Bispecific antibodies have shown promise in mitigating this issue. Interestingly, some studies suggest that antigen shedding might enhance ADC distribution in solid tumors by reducing the binding site barrier (BSB) effect, promoting more uniform drug delivery. These findings challenge the traditional emphasis on strict internalization and minimal shedding, highlighting the complexities of ADC development and the need for nuanced approaches [9].

Despite their promise, ADCs continue to face notable challenges, including dose-limiting toxicities and off-target effects. The selection of linker technology and payload remain critical to optimizing the therapeutic index. Advances in conjugation methods and antigen identification have driven the development of safer and more effective ADCs. Innovative conjugation techniques aim to enhance payload stability and homogeneity of attachment, thereby improving therapeutic outcomes [24]. Several ongoing clinical trials are evaluating ADC candidates featuring novel payloads and linker technologies to broaden their applicability across various cancer types [21,24].

## 3. Chemical Aspects of ADCs

The development of ADCs is more complex and challenging compared to nonconjugated mAbs; the components of the selected antibody for ADC generation, as well as the structure and design of the ADCs, influence their function and therapeutic efficacy [26]. This section will elaborate more on ADC engineering, chemistry, and the types of linkers used to generate different therapeutic ADCs.

### 3.1. Antibody–Drug Conjugate (ADC) Engineering

Similar to non-conjugated mAbs that are currently used for therapeutic purposes, the IgG1 isotype is commonly used to initiate ADC-dependent ADCC [26]. The IgG2 isotype may be more attractive in this therapeutic approach due to its favorable features, which include the ability to bind to four disulfide bridges compared to two found in IgG1 or IgG4 isotypes, hence enhancing its cytotoxic payload and therapeutic effect [26,27]. Moreover, IgG2 tends to form covalent dimers, which increases the binding affinity of the antibody as well as its internalization by the targeted cell [26]. Although the IgG3 isotype exhibits potent tumor-cell-killing capacity, its clinical utility in ADCs is limited due to a shortened serum half-life and stability concerns, which are also observed with the IgG4 isotype. These findings underline the critical role of antibody isotype selection in determining ADC therapeutic performance. Optimal isotype choice can enhance pharmacokinetics, stability, and effector functions, thereby improving clinical outcomes. In addition to isotype considerations, the target antigen must be predominantly expressed on tumor cells to minimize off-target toxicity. The ideal antigen should facilitate efficient receptor-mediated endocytosis of the ADC and enable effective intracellular release of the cytotoxic payload within the target cell compartment [10,28].

### 3.2. Linker Technology

Central to the therapeutic efficacy and safety of ADCs is the design of linker chemistry, which controls both systemic stability and the selective release of intracellular payload [15]. Linkers work as a bridge to connect two major components in the ADC’s structure. The linkers provide a functional handle that allows for sufficient conjugation to the antibody via a covalent bond on one side and to the cytotoxic drug on the other side of the linker. Furthermore, linkers play a crucial role in the functionality of ADCs by increasing their presence and stability in the bloodstream, providing sustained drug release at the target site [28,29,30]. An ideal linker must remain stable in circulation, resisting enzymatic degradation and hydrolysis while enabling cleavage within the tumor microenvironment or intracellular compartments [31]. Three major linker strategies have emerged: pH-sensitive, enzyme-cleavable, and non-cleavable [32]. The pH-sensitive linkers, such as hydrazones, are stable at physiological pH (~7.4) but hydrolyze in acidic endosomal/lysosomal environments (pH 4.5~6.0), as represented by gemtuzumab ozogamicin, which releases N-acetyl-γ-calicheamicin [32].

Enzyme-cleavable linkers, such as the widely used valine-citrulline (Val-Cit) linker in brentuximab vedotin and polatuzumab vedotin, exploit tumor-associated proteases like cathepsin B. These linkers are cleaved in lysosomes to release MMAE, often aided by self-destructible spacers, such as para-aminobenzyl carbamate (PABC) [32]. Additional variants include β-glucuronide linkers cleaved in necrotic regions and peptide linkers, such as Gly-Gly-Phe-Gly (GGFG), as seen in trastuzumab deruxtecan [32]. In contrast, non-cleavable linkers such as the thioether-based maleimidomethyl cyclohexane-1-carboxylate (MCC) used in trastuzumab emtansine (T-DM1), rely on complete lysosomal degradation of the antibody to release stable payload-linker amino acid complexes (e.g., lysine-MCC-DM1), offering superior plasma stability for payloads with retained activity in conjugated form [33]. In addition, advances in site-specific conjugation, such as engineered cysteine residues and enzymatic platforms (e.g., WuXiDAR4™, IDconnect™), have further improved the DAR homogeneity, enhancing the pharmacokinetics and the therapeutic index [33]. These innovations in linker architecture and conjugation chemistry have significantly expanded the therapeutic window of ADCs by enhancing tumor selectivity, reducing systemic toxicity, and facilitating the development of next-generation constructs with improved clinical performance [33].

Functionalizing amines in linkers is important for site-specific conjugation and controlled drug release. N-hydroxysuccinimide (NHS) esters coupled with ε-lysine residues introduced into an antibody scaffold are commonly used in ADC development [34]. The thiol-reactive maleimide conjugated with cysteines forms approximately 70% of the ADC structures in clinical trials [34]. However, they still suffer from chemical instability in plasma, resulting in the premature loss of the drug from the ADC, which remains to be solved [33]. Aldehyde and keto functional groups react with hydrazides and alkoxyamines to form hydrazones and oximes, which can be utilized in various applications, including bioconjugation and the development of drug delivery systems [29,35].

There are two main types of linkers commonly used for ADC development: cleavable and non-cleavable linkers. Each type has different impacts on pharmacokinetic properties, selectivity, and therapeutic index as well as the success of the ADCs [36]. For the non-cleavable linkers, there is no chemical built to cleave the linker. Once the antibody is in the lysosome, proteolytic enzymes break down the structure into its constituent amino acids, and the payload is released, containing the drug, the linker, and an amino acid [37]. This type of linker has shown high efficacy in certain tumor models; for example, ado-trastuzumab emtansine (Kadcyla^®^), which conjugates the anti-HER2 antibody trastuzumab to the maytansinoid DM1 via a non-cleavable thioether linker, is approved for the treatment of HER2-positive breast cancer. However, a notable limitation of non-cleavable linkers is their inability to mediate bystander killing [37,38]. Upon intracellular degradation of the antibody, the resulting charged drug–amino acid catabolite (e.g., lysine-DM1) is membrane-impermeable, preventing its diffusion into neighboring antigen-negative tumor cells [37,38].

Thus, ADCs designed with non-cleavable linkers might be limited to tumors expressing high tumor antigens or for treating hematological cancers [37]. On the other hand, cleavable linkers can be cleaved by specific proteases or under specific pH conditions, resulting in the release of the drug, which can subsequently kill targeted cells directly and diffuse from the cell to initiate bystander killing depending on the drug type and its physicochemical properties [38]. Although it is suggested that ADCs induce cell killing after they have been internalized by the antigen expressed on the cell surface, new evidence indicates that ADCs can initiate killing by other mechanisms, depending on the antibody, linker, and drug type [38].

Cleavable linkers are designed to release the payload selectively within the tumor microenvironment by responding to specific conditions such as acidic pH, enzymatic activity, or intracellular thiols. Examples include disulfide linkers, which are cleaved through thiol exchange; hydrazone linkers, which degrade in acidic environments; and peptide linkers, which are hydrolyzed by lysosomal proteases such as cathepsin B. While cleavable linkers enhance specificity and payload delivery to tumor cells, they pose the risk of premature cleavage in the bloodstream [23,25].

In contrast, non-cleavable linkers release their payload only after the complete degradation of the mAb within lysosomes. This mechanism offers advantages such as increased plasma stability, reduced off-target toxicity, and improved pharmacokinetics. Maleimide-based linkers are a prominent example, as they minimize premature drug release and extend ADC half-life in circulation [25].

The efficacy of ADCs is also influenced by factors such as the DAR and payload homogeneity. An optimal DAR is essential for maximizing therapeutic potency while minimizing systemic toxicity. Achieving uniformity in payload attachment ensures consistent therapeutic outcomes. Variability in conjugation can result in unpredictable pharmacological behavior, making precise and homogeneous attachment critical for ADC effectiveness [21].

Recent advances in linker technologies have introduced innovative strategies to address existing limitations. Site-specific linkers improve payload homogeneity by targeting defined attachment sites on antibodies, ensuring controlled and uniform conjugation. Tripeptide linkers enhance plasma stability and provide selective payload release within tumor cells. Additionally, hydrophilic linkers reduce ADC aggregation and improve solubility, thereby enhancing pharmacokinetics and therapeutic efficacy. Another significant innovation is the use of bioorthogonal linkers, which enable highly controlled drug release without interfering with normal cellular processes, representing a transformative step forward in ADC linker design [9].

Linkers play a critical role in optimizing ADC performance by influencing stability, selectivity, and the therapeutic index of the drug. Ongoing research into novel linker classes aims to address current challenges in toxicity, stability, and therapeutic consistency, with a focus on improving specificity and targeted drug delivery. These advancements are crucial for establishing ADCs as a cornerstone of targeted cancer therapies, providing enhanced clinical outcomes while mitigating the risks associated with off-target effects [23,25].

Despite the type of linker used to treat a specific disease, the linker’s location on the antibody should be highly considered, since it could significantly affect the pharmacokinetic and pharmacodynamic properties of the ADC and thus influence the treatment efficacy of the designed ADCs [28,39]. For example, linkers located at or near the Fab region of the antibody could interfere with or reduce the binding affinity of the ADCs with the targeted molecule. Linkers conjugated near or on the Fc area can negatively affect Fc receptor binding to immune cells or modify the antibody folding structure [10]. Overall, the type of linker, whether it is cleavable or non-cleavable, along with its location, is an essential factor that should be carefully studied before designing potent ADCs to bind and eliminate targeted cells specifically.

## 4. Conjugation Methods

ADC strategies can be broadly categorized into two main types: traditional (non-site-specific) conjugation and site-specific conjugation [5]. Both approaches involve the formation of covalent bonds between the monoclonal antibody and the cytotoxic payload, typically through reactive amino acid residues exposed on the antibody surface, most notably lysines and cysteines [5]. Precise control over the conjugation site and DAR is critical for optimizing therapeutic efficacy while minimizing off-target toxicity. Traditional lysine-based conjugation remains one of the most employed non-site-specific methods, wherein the ε-amine groups of lysine side chains react with electrophilic linkers or payloads [40]. However, this approach often leads to heterogeneous conjugates due to the random distribution of lysine residues on the antibody [40].

In many cases, the antibody or payload is first modified with thiol- or cathepsin-cleavable linkers such as valine-citrulline, which subsequently conjugate with the available lysine residues, resulting in a variable and less controlled ADC product [41]. This conjugation strategy yields a heterogeneous population of ADCs, characterized by variable DARs resulting from the random attachment of payloads to multiple lysine residues on the ADC [40]. Therefore, individual antibody molecules may carry differing numbers of cytotoxic agents, leading to batch-to-batch variability. This heterogeneity can significantly influence the pharmacokinetic profile, biodistribution, and therapeutic efficacy of the ADC, potentially affecting both safety and clinical performance [41].

### 4.1. Traditional Conjugation (Lysine-Based Conjugation)

Lysine-based conjugation is a widely used method for synthesizing ADCs, leveraging the nucleophilic ε-amino (–NH_2_) groups of solvent-accessible lysine residues. These amine groups readily react under near-neutral pH conditions, making them suitable targets for coupling with electrophilic linkers or activated drug moieties [42]. This approach enables efficient covalent attachment of cytotoxic payloads to the antibody but often results in a heterogeneous product due to the abundance and uneven distribution of lysine residues on the antibody surface [43]. Electrophilic reagents predominantly target the nucleophilic ε-amino groups of lysine residues, facilitating the covalent attachment of linker–payload constructs without requiring prior modification of the antibody backbone [42]. The clinical success of lysine-based conjugation is demonstrated by five commercially approved ADCs, highlighting the method’s utility and effectiveness [16]. This has encouraged the development of a variety of electrophilic linkers aimed at improving conjugation efficiency and stability, including NHS esters and their derivatives, benzoyl fluoride, isothiocyanates, and squaramate esters [44,45]. These linkers exhibit high reactivity toward nucleophilic heteroatoms; however, their susceptibility to hydrolysis in aqueous environments, particularly in the case of NHS esters, can significantly impact the linker–payload (LP) equivalency and overall conjugation yield [46].

Under mild reaction conditions, when lysine conjugation is delayed or inefficient, less reactive linkers may also engage with thiol groups of cysteine residues [44,45]. For instance, sulfonyl groups can selectively react with cysteine, and sulfonyl acrylates have been employed to bridge two cysteine residues, thereby functioning as bifunctional cross-linkers in ADC design [44,45]. Initially, lysine-based conjugation was considered to produce random and variable modifications due to the high reactivity of lysine-targeting linkers [47]. However, studies by Liu et al. demonstrated that the distribution of conjugation sites remains consistent across multiple batches of the same ADC when manufacturing conditions are held constant [47]. This finding suggests that, despite the potential for heterogeneity, lysine conjugation can yield reproducible site occupancy patterns under controlled production parameters, thus supporting its feasibility for large-scale and clinical-grade ADC manufacturing [48].

Variations in reaction conditions during lysine-based conjugation can also lead to heterogeneous DARs, which are associated with increased systemic toxicity and accelerated clearance. A prominent example is Mylotarg^®^ (gemtuzumab ozogamicin), withdrawn from the market due to safety concerns, including a high rate of early mortality attributed mainly to its broad DAR distribution, unstable hydrazone linker, and the use of the ultra-potent payload calicheamicin [49,50]. The ADC was re-approved in 2017 with a refined patient population, a reduced recommended dose, and a modified dosing schedule [51]. The failure of Mylotarg^®^ significantly influenced the design of subsequent lysine-conjugated ADCs. In contrast, Kadcyla^®^ (T-DM1) and Elahere^®^ (mirvetuximab soravtansine) incorporated more stable linker systems and less toxic maytansinoid payloads (e.g., DM1, DM4), resulting in higher maximum tolerated doses (MTDs) and improved therapeutic indices [51]. Under mild conjugation conditions, less reactive electrophilic linkers preferentially react with the most nucleophilic lysine residues on the mAb surface [49,50]. For example, β-lactam and methylsulfone linkers have been employed to selectively conjugate at the K99 residue within a hydrophobic pocket of the mAb, enhancing regioselectivity and stability [42].

Additionally, the K-lock™ platform enables site-specific modification at the K188 residue of the κ light chain using pentafluorophenol (PFP) esters, providing improved control over conjugation site and drug loading. These strategies contribute to greater homogeneity, batch-to-batch reproducibility, and optimized pharmacokinetic profiles in ADC manufacturing [49,50]. These site-specific lysine conjugation strategies have enabled the development of more homogeneous and stable ADCs, facilitating their advancement into clinical trials. Notably, ADCs such as A166, ZV0203, and AMT-151 have utilized these optimized conjugation approaches to improve therapeutic efficacy, safety profiles, and pharmacokinetic behavior in ongoing clinical studies [52]. The phospha-Mannich reaction has been employed to achieve site-specific modification at the K183 residue within the Fab region of trastuzumab. This chemoselective strategy enables targeted conjugation without disrupting the antigen-binding site, offering improved control over DAR and preserving the antibody’s functional integrity, making it a valuable tool for generating homogeneous ADCs with defined pharmacological properties [53]. The ε-amino (–NH_2_) group of lysine is susceptible to its local chemical environment on the antibody surface, including factors such as counter-ions, solvation effects, and neighboring functional groups. Despite the inherent variability, lysine-based conjugation remains a robust and well-established method for ADC synthesis, yielding chemically stable and reproducible products [34].

Figure 2 illustrates the two primary conjugation strategies used in ADC synthesis: (A) cysteine conjugation, in which interchain disulfide bonds are selectively reduced to generate thiol groups that react with maleimide-functionalized drug-linkers to form stable thioether bonds, offering site-selective modification, improved DAR control, and reduced heterogeneity, and (B) lysine conjugation, where ε-amino groups of solvent-exposed lysine react with activated esters (e.g., NHS esters) to yield stable amide bonds.

### 4.2. Site-Specific Conjugation (Click Reactions)

Enzymatic conjugation is a sophisticated technique that utilizes enzymes to covalently attach payloads to antibodies at specific amino acid sequences, yielding highly homogeneous ADCs with defined conjugation sites [54]. This strategy has demonstrated its effectiveness as a coupling method by consistently producing uniform ADC populations. However, many enzymatic platforms require modification of the antibody sequence or structure to introduce compatible recognition motifs [55,56]. For example, Sortase A (SrtA) is a 30 kDa transpeptidase widely applied in ADC development; it cleaves the LPXTG motif to form a thioester acyl-enzyme intermediate, which subsequently transfers the peptide–payload conjugate (peptide–LPXT) to the N-terminus of a glycine-containing acceptor substrate [55,56]. A clinically relevant example of enzyme-mediated conjugation is TRPH-222, an ADC currently in Phase I clinical trials, which utilizes the SMARtag™ platform based on formylglycine-generating enzyme (FGE) technology to introduce aldehyde tags for site-specific payload attachment [55,56].

The FGE technology specifically recognizes CXPXR motifs. It catalyzes the oxidation of the cysteine residue within this sequence to formylglycine, introducing a unique aldehyde functional group that enables site-specific bioconjugation [56]. This enzymatic strategy facilitates precise and reproducible conjugation, enhancing the homogeneity and pharmacological consistency of ADCs. Another enzyme-based approach employs protein farnesyltransferase (PFTase), which attaches isoprenoid lipid groups to cysteine residues within a CaaX box motif (“C” is cysteine, “a” is an aliphatic amino acid, and “X” is a variable residue), allowing for further chemoselective modifications [56]. A clinical example utilizing this technology is FS-1502, which was developed using the ConjuALL™ (Conjugation using Allylic Lipid Linker) platform and is currently in Phase III clinical trials [57]. Together, FGE- and PFTase-mediated bioconjugation represent powerful enzyme-driven methods for generating next-generation ADCs with improved site specificity, batch reproducibility, and therapeutic index [56].

Several enzyme-based conjugation systems are currently under investigation for ADC development in preclinical studies. These include peptide asparaginyl ligases, tubulin-tyrosine ligases, trypsiligase, phosphopantetheinyl transferases, SpyLigase, and O^6^-alkylguanine-DNA alkyltransferase, commonly known as SNAP-tag [58]. These enzymes offer diverse mechanisms for site-specific payload attachment by recognizing unique peptide sequences or chemical motifs [59]. Among the more advanced enzymatic platforms, microbial transglutaminase (mTG) catalyzes the formation of isopeptide bonds between the γ-carboxyamide group of glutamine residues and primary amines [58]. In the context of ADCs, mTG selectively targets specific native sites on antibodies, such as the Q295 residue on the Fc region [59]. A key clinical example is DP303, currently in Phase III trials, which utilizes mTG to facilitate efficient conjugation by enabling the free amine group of the payload to react with the Q295 site, resulting in highly stable and homogeneous ADCs with favorable pharmacokinetic and therapeutic properties [58]. Successful development of enzyme-conjugated ADCs requires careful optimization of both the pharmacokinetic properties of the final product and the scalability and reproducibility of the conjugation process.

Click chemistry offers numerous advantages for ADC synthesis, including mild reaction conditions, high yields, compatibility with aqueous and organic solvents, stereoselectivity, and straightforward product isolation, making it highly suitable for bioconjugation applications [60]. As a result, click reactions have become a powerful and efficient tool for bioconjugation, the process of covalently linking chemically distinct molecules, due to their versatility, reliability, and compatibility with complex biological systems [61]. Not all click reactions are inherently suitable for biological systems; for optimal bioconjugation outcomes, the reaction must be bioorthogonal to exhibit minimal or no reactivity with endogenous functional groups to preserve the native biological environment [62]. In 2002, Tornoe, Christensen, and Meldal were the first to apply click chemistry for the bioconjugation of two biological molecules utilizing solid-phase synthesis to generate peptidotriazoles through regiospecific copper(I)-catalyzed azide–alkyne cycloaddition (CuAAC) of terminal alkynes with azides, marking a pivotal advancement in site-selective biomolecular coupling [63]. Bioconjugation has become a critical tool across various scientific disciplines, including diagnostics, therapeutics, and materials science. In the context of ADC development, click chemistry can enable efficient and selective coupling of biologically active components [63,64]. Key click-based strategies employed in ADC synthesis are numerous and include the thiol-maleimide Michael addition, copper-catalyzed azide–alkyne (3+2) cycloaddition (CuAAC), oxime bond formation, the normal Diels–Alder reaction, hydrazino-iso-Pictet–Spengler (HIPS) ligation, and the inverse electron-demand Diels–Alder (IEDDA) reaction, with each offering unique advantages in terms of reaction specificity, bioorthogonality, and conjugate stability [63,64].

In ADC development, click chemistry reactions such as CuAAC, oxime formation, and the Diels–Alder reaction offer versatile, bioorthogonal, and efficient strategies for stable and site-specific bioconjugation, as detailed in the following sections.

#### 4.2.1. Copper-Catalyzed Cycloaddition of Azides and Alkynes

The first reaction formally recognized as a “click” reaction was the (3+2) cycloaddition, specifically the CuAAC, which enabled the efficient and regioselective synthesis of 1,2,3-triazoles, which have since become a foundational transformation in organic synthesis and bioconjugation chemistry [63]. This reaction is highly versatile and has been successfully applied to the modification of peptides, natural products, small-molecule drugs, DNA, and nucleotides, as well as in polymer chemistry, nanostructure fabrication and functionalization, and diverse bioconjugation strategies [65]. Azides and alkynes exhibit minimal reactivity with endogenous biological molecules due to their weak acidic and basic properties, making them essentially inert in native biochemical processes [61]. However, their inherent nucleophilic and electrophilic character allows for amenable incorporation into organic structures, enabling selective and bioorthogonal chemical modifications essential for applications in bioconjugation and chemical biology [65].

In the reaction, the copper (I) ions catalyze the selective transformation of alkynes with azides or other 1,3-dipolar compounds to form stable triazole linkages, a process characterized by high regioselectivity and reaction rates comparable to those observed in cysteine–maleimide conjugation [61]. This efficiency and specificity make CuAAC a valuable tool for bioconjugation applications due to its rapid kinetics, high specificity, bioorthogonality, and excellent yield, making it an ideal tool for selective molecular labeling and modification within complex biological environments [64]. In ADC development, CuAAC has been applied in platforms such as STRO-001, a site-specific ADC targeting CD74 developed by Sutro Biopharma, which uses click chemistry for precise payload attachment, and in dual-click strategies described by Godwin et al., where pyridazinedione linkers introduced into native disulfide bonds enable orthogonal CuAAC conjugation to generate multifunctional, homogeneous ADCs with retained antibody function [66,67]. However, a significant limitation of this reaction is the generation of reactive oxygen species (ROS) mediated by free Cu (I) ions, which can induce cytotoxicity and oxidative damage, hence limiting its applicability, particularly in in vivo settings [68]. To address this cytotoxicity associated with free Cu (I) ions in CuAAC, several mitigation strategies are under investigation, including the use of water-soluble stabilizing ligands such as tris (hydroxypropyltriazolyl)methylamine (THPTA). These ligands coordinate with Cu (I), enhancing its catalytic efficiency while reducing the generation of ROS, thereby improving the biocompatibility of the reaction for in vivo applications [63,69,70].

#### 4.2.2. Oxime Formation

The term “oxime” was first introduced by Meyer and Janny in 1882, describing a functional group formed through the proton-catalyzed nucleophilic attack of a hydroxylamine nitrogen on an electrophilic carbonyl carbon, resulting in the formation of a stable C=N–OH oxime bond [35]. This reaction has since become a valuable tool in bioconjugation due to its selectivity and compatibility with aqueous environments [35]. The oxime bond exhibits significantly greater stability in aqueous environments and at physiological pH compared to analogous linkages such as hydrazones or imines, making it a more favorable and reliable option for bioconjugation applications, particularly in biological systems where hydrolytic resistance is essential [71]. The oxime ligation reaction is both effective and chemoselective, offering high specificity for carbonyl-containing substrates; however, oxime bond formation proceeds more slowly under neutral pH conditions compared to other click reactions, which may limit its kinetics in specific biological applications unless catalyzed or performed under mildly acidic conditions [72].

Aniline was initially introduced as a catalyst for oxime ligation due to its good solubility and nucleophilic inertness under physiological conditions [73]. Its presence significantly accelerates the rate of oxime bond formation, with studies showing that aniline can enhance the reaction rate by up to 40-fold at neutral pH, thus improving the efficiency of bioconjugation under biologically relevant conditions [74]. Nevertheless, due to its cytotoxicity, aniline is not suitable for use in bioconjugation reactions involving live cells [73]. Consequently, other alternative catalysts that can accelerate oxime bond formation while maintaining low toxicity are explored [75].

Compounds such as anthranilic acids, aminobenzoic acids, 2-aminophenols, and 2-(aminomethyl)-benzimidazoles have been investigated as alternative catalysts for oxime ligation [73,76]. These catalysts not only exhibit improved biocompatibility but also accelerate the reaction rate by at least twofold compared to aniline under neutral pH conditions, making them promising candidates for efficient and non-toxic bioconjugation in biological systems [76,77]. Notable examples of ADCs developed using oxime ligation include ARX788 and AGS62P1/ASP1235, which employ site-specific incorporation of *p*-acetylphenylalanine (*p*-AcF) into the antibody, enabling conjugation to aminooxy-functionalized payloads, and FS-1502/LCB14–0110, developed by LegoChem, which uses a C-terminal CAAX tag and enzymatic modification to install ketone handles for subsequent oxime-based linker attachment, resulting in homogeneous, stable ADCs with improved pharmacokinetics and therapeutic performance [78,79].

#### 4.2.3. Diels–Alder Reaction (DA)

The Diels–Alder (DA) reaction, first described by Otto Diels and Kurt Alder in 1928, is a (4+2) cycloaddition reaction between a conjugated diene and a dienophile, forming a six-membered ring. This reaction offers numerous advantages, including high regio- and stereoselectivity, mild reaction conditions, tolerance to a wide range of functional groups, and the ability to proceed without the need for catalysts, making it a valuable tool in synthetic chemistry, materials science, and bioconjugation [80]. The DA reaction can be performed without the use of costly or hazardous reagents or solvents, enhancing its practicality and environmental compatibility. Readily available dienes and dienophiles such as furan, maleimide, naphthalene, and anthracene facilitate broad applicability [81,82]. Notably, the DA reaction is compatible with aqueous environments, where it often proceeds more rapidly due to enhanced hydrogen bonding between the activated complex and reactants, as well as stronger hydrophobic interactions that promote effective molecular collisions, further increasing reaction efficiency in water [81,82]. The DA reaction does not require stringent temperature control and can proceed effectively at room temperature, albeit with reduced kinetic rates. Despite this, product formation remains feasible under mild conditions.

Owing to its inherent stereoselectivity, high yields, ease of purification, and minimal generation of toxic by-products, the DA reaction aligns with the principles of click chemistry, making it a desirable strategy for bioconjugation [83]. The classical DA reaction has been effectively applied in the synthesis of ADCs. One such approach utilizes electron-rich dienes, such as furan and various cyclodiene derivatives, to enable site-specific modification of antibodies [84]. The cytotoxic payload is subsequently linked via maleimide-based dienophiles [84]. In a study by Amant et al., trastuzumab was employed as the antibody scaffold and vedotin as the payload, both of which are components of FDA-approved ADCs [85]. In this method, lysine residues on the antibody were selectively modified using NHS-activated esters, converting them into cyclopentadiene-functionalized sites. The DA cycloaddition with maleimide-modified vedotin was performed at room temperature for 4 h, yielding high product efficiency [85]. Both in vitro and in vivo evaluations demonstrated that the resulting linkers exhibited greater stability than conventional thiol–maleimide linkers while maintaining potent cytotoxicity [85]. Importantly, this DA-based conjugation strategy avoids many of the technical limitations associated with thiol-maleimide chemistry, offering a more robust and scalable route for ADC development [84].

In addition to strategies that modify naturally occurring residues, click chemistry can be integrated with genetically encoded unnatural amino acids (uAAs) to achieve precise and site-specific payload attachment [86]. This approach introduces orthogonal reactive groups such as *p*-acetylphenylalanine, azido-lysine, or strained alkynes into predetermined positions within the antibody during biosynthesis, enabling bioorthogonal reactions without interference from native amino acids [86]. By providing absolute control over the conjugation site, uAA technology generates highly homogeneous ADCs with defined DARs, reduced batch variability, and predictable pharmacokinetic and pharmacodynamic profiles [87]. Moreover, site selection can be optimized to preserve antigen-binding affinity, minimize steric hindrance, and enhance linker stability, thereby improving the therapeutic index and tumor penetration [88]. Clinically relevant examples include ARX305, which demonstrates the potential of uAA-enabled click chemistry to deliver stable, site-specific, and pharmacologically optimized ADCs [87,88].

Figure 3 summarizes the three representative click chemistry reactions applied in ADC development. (A) The CuAAC enables the formation of stable triazole linkages between azide- and alkyne-functionalized antibody and payload components. (B) Oxime bond formation involves the chemoselective reaction of an aminooxy group with a carbonyl moiety, generating a hydrolytically stable C=N–OH linkage. (C) The DA reaction achieves site-specific conjugation through a (4+2) cycloaddition between a diene-functionalized antibody and a dienophile-linked payload, producing a stable six-membered ring. Table 2 summarizes the primary conjugation methods used in ADCs, detailing their chemical basis, type, advantages, limitations, representative examples, and supporting references from the literature.

## 5. Tailoring ADCs to Specific Diseases

### 5.1. Tailoring ADCs to Cancer

Numerous ADCs have been developed for both hematological malignancies and solid tumors. Among them, those targeting HER2 in metastatic breast cancer are the most widely used [16]. The success of approved ADCs is primarily attributed to the diversification of target antigens and advances in payload chemistry, which have generated significant commercial interest in developing novel ADCs [89].

In hematological cancers, there is one ADC for AML and five others for B-cell malignancies [90]. As alternative therapies, such as bispecific antibodies, unconjugated monoclonal antibodies, and chimeric antigen receptor (CAR) T cells, have demonstrated remarkable success in these settings, ADCs must offer clear clinical benefits to remain competitive [91]. Gemtuzumab ozogamicin was approved again in 2017 for the treatment of CD33-positive relapsed or refractory AML in adults and children [2,89]. Other strategies include Moxetumomab Pasudotox (targeting CD22) and ricin-based fusion proteins targeting CD3 and CD7, though these were ultimately withdrawn due to toxicity, immunogenicity, or association with graft-versus-host disease [92].

Notably, Brentuximab Vedotin, an ADC targeting CD30, has become a key treatment for Stage III and IV Hodgkin’s lymphoma [93]. In adult patients with previously untreated disease, a combination of Brentuximab Vedotin with doxorubicin, vinblastine, and dacarbazine (A+AVD) demonstrated a six-year overall survival rate of 93.9% in long-term follow-up [94]. Compared to the traditional ABVD regimen (doxorubicin, bleomycin, vinblastine, dacarbazine), the A+AVD group had fewer instances of second cancers and a reduced need for subsequent therapy [94]. In pediatric patients with advanced-stage disease (IIB–IVB), Brentuximab Vedotin combined with multi-agent chemotherapy yielded a three-year event-free survival of 92.1% and an overall survival of 99.3% [95], supporting its benefit across age groups. These clinical results reinforce CD30 as a validated hematologic target and highlight the continued evolution of ADCs alongside other targeted therapies. ADC targets validated in other hematologic malignancies include CD22, CD79b, CD19, and B-cell maturation antigen (BCMA) [2,89].

In solid tumors, HER2-positive breast cancer remains the most common target of ADCs, followed by other HER2-expressing tumors such as non-small cell lung cancer (NSCLC) and urothelial carcinoma [96]. For example, MRG002, developed by Shanghai Miracogen, is under investigation for HER2-positive NSCLC and bladder cancer [96]. Additionally, Enfortumab Vedotin, which targets Nectin-4 in urothelial malignancies, and Sacituzumab Govitecan, which targets TROP-2 in TNBC, represent key examples of promising ADCs in solid tumors [97]. Folate receptor alpha (FRα) is another antigen expressed across multiple tumor types. Mirvetuximab Soravtansine, developed for FRα-positive patients, has shown potential in treating peritoneal, ovarian, and fallopian tube cancers [98]. It received accelerated FDA approval in 2022. In clinical studies involving patients with high FRα expression, an overall response rate of 32.4% was reported. Other FRα-targeting ADCs, such as Luveltamab and Tazevibulin, are also in clinical evaluation for these indications [2,89].

HER2 is frequently overexpressed in multiple cancers (including breast cancer) and is associated with poor prognosis and increased risk of metastasis. Its extracellular domain (ECD) is accessible on the cell surface, making it a prime target for antibody-based therapies [99]. Trastuzumab was the first monoclonal antibody approved for HER2-positive metastatic breast cancer. It binds to subdomain IV of the HER2 ECD, inhibiting downstream HER2-HER3-phosphoinositide 3-kinase (PI3K) signaling, preventing ECD shedding and promoting antibody-dependent cellular cytotoxicity (ADCC) through engagement of the fragment crystallizable (Fc) region [99].

Trastuzumab Deruxtecan (T-Dxd) is a next-generation ADC composed of Trastuzumab linked to a topoisomerase I inhibitor via a cleavable peptide linker. It has been approved for HER2-positive metastatic breast and gastric cancers. DHES0815A is an investigational HER2-targeting ADC comprising a humanized HER2 antibody (HER2-THIOMAB) that binds subdomain I of the HER2 ECD [100]. It is conjugated to a modified PBD payload via a hindered disulfide linker and light chain amino acid engineering for improved stability [100]. Although preclinical results were promising, the Phase I clinical trial was terminated due to safety concerns [2,101].

Trastuzumab Emtansine (Kadcyla, T-DM1) is another HER2-targeted ADC in a Phase III trial involving patients with HER2-positive early breast cancer and residual invasive disease following neoadjuvant systemic therapy. T-DM1 significantly improved outcomes compared to Trastuzumab alone. After a median follow-up of 8.4 years, the seven-year invasive disease-free survival rate was 80.8%, and overall survival reached 89.1%. These findings highlight the capacity of T-DM1 to reduce mortality in this high-risk population [102].

### 5.2. Tailoring ADCs for Non-Cancer Diseases

While ADCs have demonstrated significant success in oncology, their application is increasingly expanding into non-oncologic diseases, broadening the potential of targeted drug delivery systems [85]. In infectious diseases, ADCs offer a novel strategy to selectively eliminate pathogen-infected host cells by targeting antigens expressed or upregulated during infection [85]. ADCs can be engineered to target *Mycobacterium tuberculosis*-infected macrophages and deliver antimicrobial payloads directly to infected cells. For example, an anti-CCRL2 ADC carrying SG3249 enhanced bacterial clearance and reduced lung inflammation when combined with standard therapy in mice [103]. ADCs targeting the HIV envelope glycoprotein (gp120) on latently infected T cells have also demonstrated potential in removing viral reservoirs [104]. Similarly, anti-bacterial ADCs have shown bactericidal activity against *Staphylococcus aureus* (*S. aureus*), providing promising support for antibiotic therapy in the face of rising antimicrobial resistance [105].

In autoimmune diseases, ADCs are also being developed to selectively deplete pathogenic immune cell subsets using cytotoxic or immunosuppressive payloads [106]. CD79b-targeted ADCs, initially designed for B-cell malignancies, have shown efficacy in preclinical models of systemic lupus erythematosus (SLE) by depleting autoreactive B cells while sparing protective immunity [106]. Likewise, anti-CD45 ADCs conjugated with saporin have demonstrated potent immunomodulatory effects in murine models of RAG deficiency, enabling robust multilineage hematopoietic stem cell engraftment, immune reconstitution, and thymic epithelial cell recovery while minimizing the systemic toxicity associated with conventional conditioning regimens such as total body irradiation [107]. These advancements highlight the versatility of the ADC platform as a modular therapeutic system adaptable to diverse biological targets.

Regarding the use of ADCs in infectious diseases, several challenges arise in designing an effective ADC therapy. The first challenge is selecting the target due to the pathogen’s specificity. Unlike ADCs used for cancer therapy, which target tumor-associated antigens, ADCs used for infectious diseases must be designed to identify antigens on the surface of bacteria or viruses that can be easily accessed [105]. The successful selection of antigens to be targeted by ADCs requires that they be abundantly expressed on the pathogen surface [108]. For example, β-N-acetylglucosamine is highly abundant on the surface of *S. aureus*, providing approximately 50,000 binding sites [108].

Additionally, other criteria should be considered for the antigen, such as its high degree of conservation across strains, which reduces the potential for resistance emergence; it should be easily accessible during active infection; and it should be absent or minimally expressed on host tissues [108]. Solutions that can overcome this challenge include applying advanced conjugation methods to create homogeneous ADCs with defined ADR. For example, THIOMAB Technology is a homogeneous ADC utilizing engineered cysteine residues for precise linker attachment by introducing cysteine residues in the heavy or light chains of antibodies at specific positions [109]. A case study was performed to overcome obstacles in this first challenge utilizing DSTA4637S. DSTA4637S is a clinically advanced ADC for infectious diseases [110]. It consists of thiomab human immunoglobulin G1, which is linked to a novel rifamycin-class compound [110]. This ADC can target the *S. aureus* wall successfully by binding to teichoic acid [110].

The second challenge is selecting the appropriate antimicrobial payload. ADCs require specific payload criteria for successful therapeutic applications, including retaining bactericidal activity at low concentrations, possessing suitable functional groups for conjugation, maintaining stability under physiological conditions, and overcoming bacterial resistance mechanisms [105,108]. This obstacle can be resolved by using a dual-payload system. For instance, VSX-AMP is an ADC that incorporates an antimicrobial peptide targeting *Pseudomonas aeruginosa*. It contains the VSX monoclonal antibody, which targets *P. aeruginosa* in the Lipopolysaccharide (LPS) core, and D297 antimicrobial peptides that are fused to the variable regions of the antibody [111,112]. VSX-AMP is highly effective as a bactericidal agent, in addition to its low cytotoxicity against mammalian cells [111,112].

One of the challenges that ADCs face is the need for specialized linker technologies with specific properties. These properties are the ability to release the drug in bacterial environments rather than mammalian cells, resistance to degradation by bacterial enzymes, and ability to maintain stability in infected tissues [105,108]. It can be performed by using a smart linker in ADCs that incorporate environment-responsive release mechanisms. For instance, a thiol can be used as a linker that can be activated through interaction with the bacterial surface. It was reported that a novel mechanism for drug release at the bacterial cell surface occurs via thiol–disulfide exchange between the thiol linker in the ADCs and extracellular thiols of the pathogen, leading to the antimicrobial efficacy of the ADC both in vitro and in vivo using an implant-associated osteomyelitis model [113].

## 6. Advances in Delivery Systems

### 6.1. Nanoparticle-Based Delivery

Antibody-functionalized nanoparticles have gained traction as a complementary or alternative strategy to ADCs, offering targeted delivery across diverse disease areas by combining antibody specificity with nanocarrier versatility [114]. Unlike ADCs, which directly link a cytotoxic payload to an antibody, these nanoparticle systems serve as carriers that encapsulate therapeutic agents and are externally functionalized with antibodies. Their key advantages include enhanced stability, tunable pharmacokinetics, and the ability to incorporate multiple therapeutic and diagnostic agents [115,116].

Among nanocarriers, liposomes are the most clinically advanced. Doxil^®^, a PEGylated liposomal formulation of doxorubicin, was the first FDA-approved nanodrug, utilizing the enhanced permeability and retention (EPR) effect for tumor targeting [117,118]. Liposomes are composed of an aqueous core that encapsulates a hydrophilic drug and one or more phospholipid bilayers, allowing for the encapsulation of hydrophobic drugs within the lipid bilayer [119]. Antibody-conjugated liposomes (e.g., trastuzumab-decorated liposomes for HER2-positive cancers) have demonstrated enhanced selective uptake and antitumor efficacy in preclinical and early clinical studies [120]. Other constructs, such as anti-EGFR and anti-c-Met-functionalized liposomes, have shown improved drug accumulation and tumor regression in murine models [121,122].

Lipid nanoparticles (LNPs), notably used in the BNT162b2 and mRNA-1273 COVID-19 vaccines, have demonstrated exceptional transfection efficiency, biodegradability, and endosomal escape capabilities, facilitating robust gene expression [123,124,125,126]. Functionalizing LNPs with antibodies targeting specific cell surface markers—such as EGFR, CD4, or PV1—has enabled organ-specific or immune cell-specific gene delivery. This has shown particular promise in cancer immunotherapy, pulmonary disease targeting, and genome editing, with studies reporting up to 40-fold increases in lung-specific protein expression and 30-fold enhancement in mRNA uptake by CD4^+^ T cells [127,128,129].

Similarly, polymeric nanoparticles (e.g., PLGA, PLA) provide sustained drug release, tunable surface modification, and excellent biocompatibility. Antibody-conjugated variants, such as trastuzumab-functionalized paclitaxel nanoparticles, have demonstrated enhanced cytotoxicity in HER2-positive cancers. Others targeting CEA or ICAM-1 have been explored for colorectal tumors and gastrointestinal inflammation, showing improved tissue specificity and retention [130,131,132,133].

Protein-based nanoparticles, including albumin and gelatin formulations, further enhance delivery through their natural biodegradability and capacity for precise antibody conjugation. Bevacizumab-coated albumin nanoparticles have shown 50% higher paclitaxel delivery in gynecologic cancers, while gelatin-based nanoparticles functionalized with anti-CD3 selectively internalized into T lymphocytes, indicating their utility in immune modulation [134,135].

Gold and metal oxide nanoparticles combine optical/magnetic properties with antibody-guided specificity for both therapeutic and diagnostic purposes. Anti-HER2 conjugated gold nanostructures enable photothermal tumor ablation, while iron oxide nanoparticles conjugated with anti-VEGF or CD105 antibodies support MRI-guided therapy and targeted cytotoxicity in HER2-positive cancers [136,137,138,139,140].

Although antibody-functionalized nanoparticles are structurally distinct from ADCs and utilize different drug delivery mechanisms, their modularity, multivalency, and capacity to carry a range of therapeutic cargos—including nucleic acids, small molecules, and proteins—make them promising alternatives in targeted therapy, especially where conventional ADCs may face biological or pharmacological limitations. Table 3 is a summary of key types of antibody-functionalized nanoparticles explored for targeted drug delivery across oncology, infectious, and immune-related diseases. It highlights target antigens, therapeutic cargos, disease indications, and notable experimental or clinical outcomes.

### 6.2. Bioconjugation Techniques

The clinical success of next-generation ADCs depends heavily on bioconjugation strategies that ensure stability, precise drug loading, and controlled payload release. Modern approaches focus on three main goals: improving serum stability, minimizing off-target toxicity, and tailoring the therapeutic index through precision engineering.

#### 6.2.1. ADC Stability

Conventional lysine- and cysteine-based conjugation approaches, while historically favored for their synthetic simplicity and scalability, often generate heterogeneous ADC populations with broad DAR distributions and multiple, non-uniform conjugation sites. Antibodies typically contain ~80–90 solvent-accessible lysine residues and eight interchain cysteine residues, many of which are chemically reactive under conjugation conditions, leading to stochastic payload attachment and DAR species ranging from 0 to 8 [153,154]. This heterogeneity alters hydrophobicity, antigen-binding affinity, and payload release kinetics, with high-DAR species prone to aggregation, accelerated hepatic clearance, and increased systemic toxicity, whereas low-DAR species may exhibit insufficient cytotoxic potency, collectively reducing the therapeutic index [153,154]. To overcome these limitations, site-specific conjugation methods have been developed to yield homogeneous ADCs with precisely defined DARs (commonly 2 or 4) and consistent payload positioning. Engineered cysteines introduced at structurally permissive sites, such as S239C in the Fc domain or L328C in the CH2 domain, enable selective thiol–maleimide coupling without disrupting native disulfide bonds, while enzymatic approaches such as microbial transglutaminase-mediated modification at Q295 in the Fc region allow covalent linker–payload attachment with minimal impact on Fc receptor binding or antigen recognition [9,41]. These precision conjugation strategies minimize DAR variability, preserve antibody folding, enhance plasma stability by reducing deconjugation rates, and improve pharmacokinetic predictability [9,41].

#### 6.2.2. Precision Conjugation Strategies

While the major linker classes have been described previously, recent developments build on these foundations by integrating tumor-selective triggers and modular designs. Examples include legumain-cleavable linkers (Vincerx Pharma) that remain stable in circulation but release payloads in tumor lysosomes and linkers incorporating cell-trapping moieties to limit payload diffusion and enhance intratumoral retention. Such refinements allow for fine-tuning of the bystander effect, biodistribution, and DAR [10,26,27,155,156].

Building upon this premise, Vincerx Pharma has created an advanced modular ADC platform that incorporates customizable linkers, innovative payloads, and targeting components. An essential innovation is their legumain-cleavable linker, which maintains stability in circulation yet is specifically cleaved in tumor lysosomes, where legumain is overexpressed. This specificity reduces systemic toxicity and improves tumor selectivity. Moreover, Vincerx’s payload, a kinesin spindle protein inhibitor (KSPi), demonstrates significant efficacy in proliferating tumor cells while preserving healthy, non-proliferating tissues [157]. The addition of a cell-trapping moiety further limits the payload’s migration into non-cancerous cells, thereby enhancing intratumoral accumulation. This comprehensive modular strategy enables the precise adjustment of the bystander impact, DAR, and therapeutic index, illustrating the capability of modular ADC engineering to enhance efficacy and safety for various oncological targets [157].

#### 6.2.3. Minimizing Off-Target Effects

Beyond stable linkers, off-target mitigation strategies include engineering payloads with reduced membrane permeability, selecting antibodies with higher tumor selectivity, and modifying Fc regions to decrease hepatic uptake. Conditional activation technologies, such as Probody ADCs with protease-cleavable masks, restrict ADC activity to the tumor microenvironment, further widening the therapeutic window [158,159]. Optimizing DAR is essential, as an increased DAR enhances systemic clearance and liver accumulation due to augmented hydrophobicity [26,160]. Antibody engineering is crucial for reducing off-target effects in ADC therapy [161]. Moreover, emerging strategies like conditionally active ADCs, particularly Probody therapies, utilize masking peptides that are exclusively cleaved by tumor-specific proteases. This guarantees the activation of the ADC only within the tumor microenvironment, thus protecting healthy organs from exposure [162].

#### 6.2.4. Outlook for Next-Generation ADCs 

The integration of site-specific conjugation strategies enabling precise DAR control, tumor-selective linker systems responsive to proteases (e.g., cathepsin B, legumain) or acidic pH, payloads with optimized physicochemical properties to modulate membrane permeability and hydrophilicity, and conditional activation platforms such as protease-activated probody ADCs is driving the development of next-generation ADCs with improved plasma stability, reduced off-target cytotoxicity, and enhanced intratumoral drug release [27,163]. Collectively, these advancements are expanding ADC applicability to heterogeneous and low-antigen-density solid tumors as well as emerging non-oncologic indications, including infectious and autoimmune diseases [31,164,165].

## 7. Challenges and Future Directions

While ADCs have achieved significant success in cancer treatment, their broader use is still hindered by several key challenges. One of the primary concerns is off-target toxicity, which occurs when ADCs bind to low levels of target antigens present on healthy cells [11]. Moreover, when the linker connecting the drug to the antibody breaks prematurely, this could lead to the release of the toxic payload into the bloodstream. Therefore, harmful effects on normal tissues and serious side effects such as neutropenia, liver damage, and nerve toxicity have been observed. As a result, the therapeutic window is narrowed, and patient safety is compromised, particularly when the targeted antigen is not exclusively expressed by cancer cells [166].

Another primary concern is tumor heterogeneity, where differences in antigen expression within the tumor or among metastatic sites can lead to irregular ADC distribution and decreased effectiveness [30]. In addition, resistance to ADC therapy is becoming increasingly recognized. Cancer cells can develop various defense mechanisms, such as shedding or mutating the target antigen, reducing antigen expression, hindering internalization of the ADC, impairing lysosomal drug release, or increasing the activity of drug-efflux transporters that pump out the cytotoxic agents [167].

The manufacturing of ADCs also presents significant challenges. Their production involves complex and tightly controlled procedures to ensure consistent DAR, stability, and biological function [85]. Ensuring product uniformity, preventing aggregation, and avoiding immune-reactive impurities are critical but technically challenging aspects of quality control. Moreover, the overall production process is expensive due to the multiple steps involved in synthesis and purification as well as the need for specialized facilities to handle toxic materials [168].

To address the current limitations of ADCs, researchers are pursuing a variety of advanced strategies that highlight the evolving landscape of ADC technology. A notable innovation is the use of site-specific conjugation methods, which enable the attachment of cytotoxic drugs to specific sites on the antibody. Techniques such as enzymatic ligation (e.g., using sortase or transglutaminase), click chemistry, and the incorporation of non-natural amino acids are being employed to create highly uniform ADCs with defined DARs, enhanced pharmacokinetics, and minimized off-target toxicity [169,170].

Another promising direction is the development of bispecific ADCs. These molecules are designed to recognize two different tumor antigens simultaneously, which enhances targeting accuracy, promotes efficient internalization, and reduces the risk of resistance due to antigen loss or variation [171]. Preliminary preclinical findings suggest that bispecific ADCs can achieve superior tumor cell killing, particularly in cancers with diverse antigen expression profiles [172].

Researchers are also working on next-generation payloads. While conventional ADCs often rely on microtubule disruptors or DNA-interacting agents, novel classes of cytotoxins, such as immune activators, RNA polymerase inhibitors, and proteolysis-targeting chimeras (PROTACs), are being explored. These innovative agents may help overcome resistance mechanisms and introduce additional therapeutic effects, such as immunogenic cell death or bystander killing [173,174].

Moreover, advances in linker technology are enhancing the precision of drug release. Linkers that respond to specific tumor conditions, such as acidic pH, high levels of glutathione, or certain tumor-associated enzymes, are being designed to release the payload only within the tumor environment, thereby limiting systemic toxicity [15]. Linking these strategies with delivery platforms, such as nanoparticles or liposomes, further enhances ADC stability, bioavailability, and tumor penetration, while potentially reducing immune system clearance.

Due to the complex nature of cancer biology and the diverse mechanisms driving drug resistance, combination therapies that include ADCs are becoming increasingly prominent in both laboratory and clinical research. An auspicious approach involves pairing ADCs with immune checkpoint inhibitors (ICIs), such as anti-PD-1 or anti-CTLA-4 antibodies. The cytotoxic effect of ADCs can lead to tumor cell death, which in turn enhances antigen presentation and stimulates immune activation, potentially transforming immunologically “cold” tumors into “hot” ones that are more responsive to immunotherapy [175].

Another path under exploration is the combination of ADCs with targeted agents, such as tyrosine kinase inhibitors (TKIs). These combinations may increase the susceptibility of cancer cells to ADCs by interfering with signaling pathways that regulate cell survival, proliferation, or resistance. For example, inhibiting the PI3K/Akt pathway may potentiate the pro-apoptotic effects of ADC payloads [176].

In the context of blood cancers, integrating ADCs into chemo-immunotherapy regimens has produced synergistic effects, resulting in improved response rates and more robust remissions. Furthermore, using ADCs in combination therapies may enable lower doses of each component, reducing toxicity while maintaining therapeutic impact [177]. However, these combination strategies require careful design to ensure compatibility in terms of pharmacodynamics, side effect profiles, and treatment scheduling. Ongoing clinical trials play a vital role in refining these regimens and identifying predictive biomarkers that can help select the patients most likely to benefit from such approaches.

## 8. Conclusions

Currently, ADCs represent a significant advancement in targeted therapeutics, combining the specificity of monoclonal antibodies with the potency of cytotoxic payloads to achieve selective tumor cell killing while minimizing systemic toxicity. Innovations in antibody engineering, linker chemistry, and site-specific conjugation have significantly enhanced the stability, efficacy, and safety of ADCs, leading to their approval in various malignancies and the development of hundreds more in clinical trials. Beyond oncology, ADCs are being explored in infectious and autoimmune diseases, demonstrating their versatility and potential for modular design. Despite challenges such as antigen heterogeneity, resistance, and manufacturing complexity, emerging strategies, including bispecific and dual-payload ADCs, immune-stimulating conjugates, and nanoparticle integration, are addressing current limitations and expanding therapeutic applications. As technological and translational advances continue, ADCs are sure to become a cornerstone of precision medicine across diverse diseases.

## Figures and Tables

**Figure 1 pharmaceutics-17-01164-f001:**
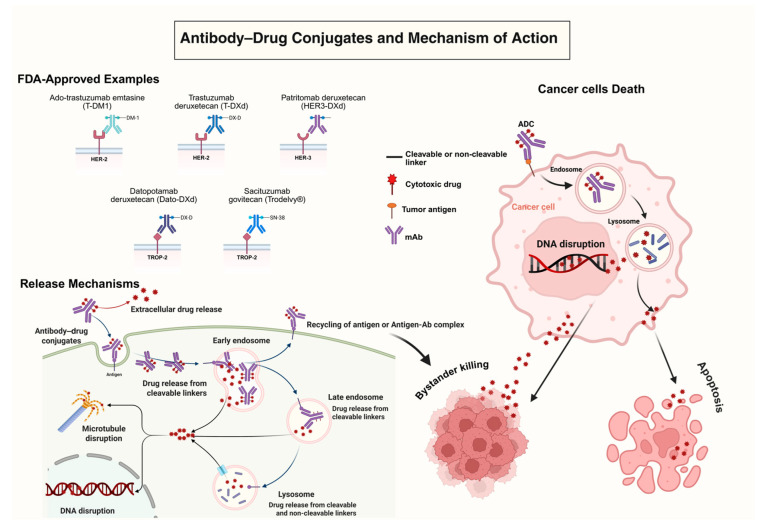
Mechanism of action and therapeutic pathways of antibody–drug conjugates (ADCs). This schematic illustrates the key components and mechanisms underlying the function of ADCs. (**Top left**), examples of FDA-approved ADCs targeting HER2 (e.g., T-DM1, T-DXd), HER3 (HER3-DXd) and TROP2 (Dato-DXd, Trodelvy^®^), each conjugated to a potent cytotoxic payload. (**Top right**), upon binding to tumor-associated antigens, ADCs are internalized into cancer cells via receptor-mediated endocytosis, followed by trafficking to the lysosome, where the cytotoxic payload is released through either cleavage of the linker or lysosomal degradation. The released drug induces cell death via mechanisms such as DNA disruption or microtubule destabilization, leading to apoptosis. (**Bottom left**), multiple release pathways are shown, including extracellular drug release, endosomal and lysosomal processing, and recycling of the antibody–antigen complex. (**Bottom right**), some payloads also exhibit a bystander effect, allowing cytotoxic diffusion to neighboring antigen-negative cells, thereby enhancing antitumor efficacy in heterogeneous tumor microenvironments. Created by Biorender.com.

**Figure 2 pharmaceutics-17-01164-f002:**
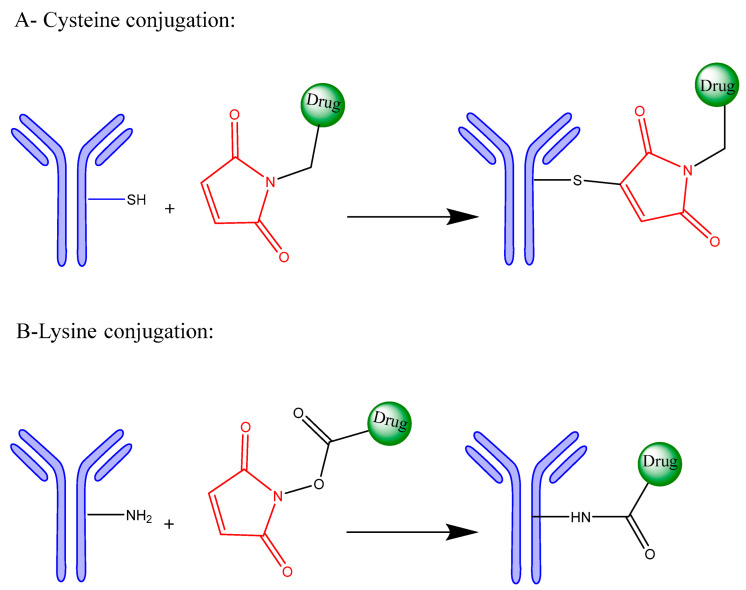
Antibody drug conjugations represent the two primary conjugation strategies in ADC synthesis. (**A**) Cysteine conjugation: reduction of interchain disulfide bonds generates free thiol groups that react with maleimide-functionalized drug-linkers to form stable thioether bonds. (**B**) Lysine conjugation: ε-amino groups of solvent-exposed lysines react with activated esters (e.g., NHS esters) of drug linkers to yield stable amide bonds.

**Figure 3 pharmaceutics-17-01164-f003:**
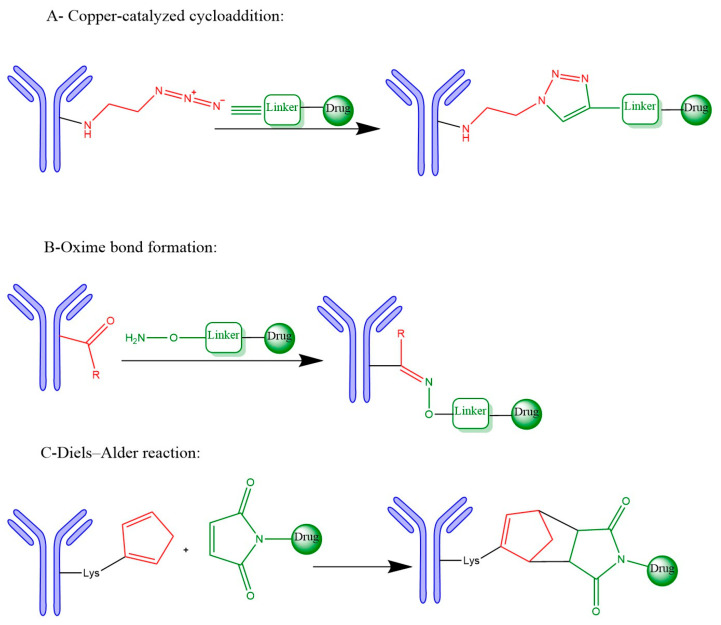
Schematic illustration of site-specific conjugation strategies for ADCs, including (**A**) copper-catalyzed azide–alkyne cycloaddition, (**B**) oxime bond formation, and (**C**) Diels–Alder cycloaddition, enabling selective and stable drug–antibody linkage.

**Table 1 pharmaceutics-17-01164-t001:** ADC’s payload examples approved for clinical market worldwide use.

Drugs	Chemical Structures	IC_50_	Approved Date
Belantamab	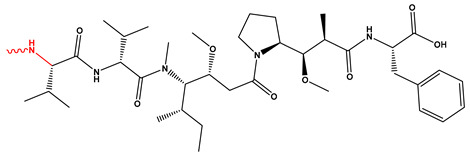	≈6.0 nM	5 August 2020
Cetuximab sarotalocan	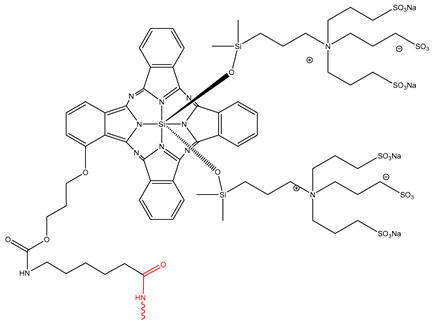	2.2–5.2 nM	25 September 2020
Fam-trastuzumab deruxtecan	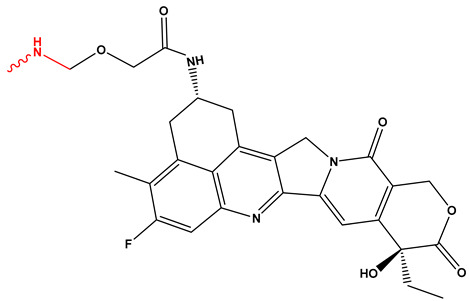	≈0.03 nM	20 December 2019
A do-trastuzumab Emtansine	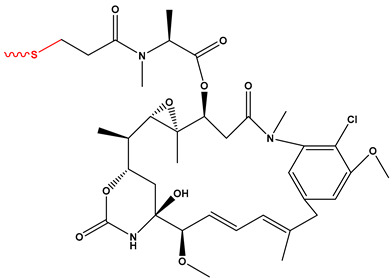	≈0.24 nM	22 February 2013
Polatuzumab vedotin	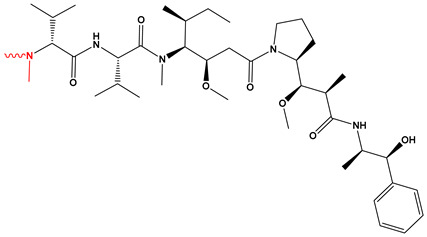	≈0.07 nM	10 June 2019
Loncastuximab tesirine	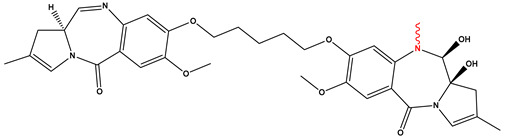	≈0.03 nM	23 April 2021
Gemtuzumab ozogamicin	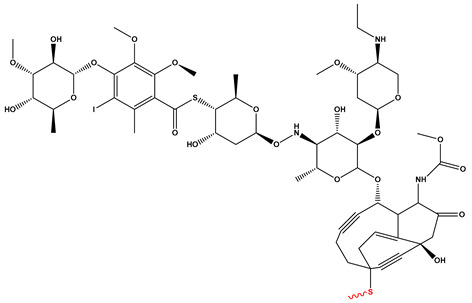	≈0.5 nM	1 September 2017

**Table 2 pharmaceutics-17-01164-t002:** Overview of ADC strategies: mechanisms, applications, and key considerations.

Conjugation Method	Type	Reactive Group	Advantages	Limitations	Examples	References
Lysine-Based Conjugation	Traditional (non-site-specific)	ε-Amine (–NH_2_) of lysine	Clinically validated; widely used; multiple FDA-approved ADCs	Heterogeneous DAR; off-target variability; batch-to-batch inconsistency	Uses NHS esters, isothiocyanates, squaramate esters; ~5 approved ADCs	[41,43,44,45]
Cysteine-Based Conjugation	Traditional and site-selective	Thiol (–SH) group of cysteine	More controlled DAR than lysine; accessible residues upon disulfide reduction	Risk of disulfide bond disruption; partial heterogeneity	Thiol–maleimide chemistry; some FDA-approved ADCs (e.g., brentuximab vedotin)	[48,49,50]
Enzyme-Mediated Conjugation	Site-specific	Engineered peptide sequences (e.g., glutamine, lysine tags)	Precise site control; minimal heterogeneity; high reproducibility	Requires antibody engineering and enzyme handling	Transglutaminase, sortase A, formylglycine-generating enzymes	[52,53,55,56]
Genetically Encoded Unnatural Amino Acids (UAAs)	Site-specific	Non-natural functional groups (e.g., azides, alkynes)	High precision; enables click chemistry; stable constructs	Requires codon reassignment and expression optimization	Azide–alkyne click; oxime ligation; stable and homogeneous ADCs	[58,62,63]
Diels–Alder Cycloaddition (DA)	Site-specific	Cyclopentadiene and maleimide (diene–dienophile pair)	Mild aqueous conditions; high yields; avoids thiol–maleimide issues	Limited in vivo validation; slower at room temperature	Furan–maleimide system; DA-based trastuzumab–vedotin synthesis	[80,81,82,84]

**Table 3 pharmaceutics-17-01164-t003:** Antibody-conjugated nanoparticle delivery systems: platforms, targets, therapeutic applications, and outcomes.

Nanoparticle Type	Antibody Target	Disease	Cargo	Outcomes	References
Liposomes	EGFR, HER2, c-Met, TfR, NS, pRBCs, E. coli, actin	Cancer (e.g., breast, lung, GBM), Alzheimer’s, malaria, bacterial infections, stroke	Doxorubicin, plasmid DNA (p53), chloroquine, fosmidomycin, PmB, TMZ	Improved targeting and reduced tumor size; 50% hemorrhage reduction; 10× drug efficacy against pRBCs; selective antibacterial delivery	[117,118,119,120,121,122,141]
Lipid Nanoparticles (LNPs)	EGFR, PV1, CD4	Cancer, lung diseases, immunotherapy	mRNA, gene constructs	40× lung-specific protein expression; 30× mRNA uptake in CD4^+^ T cells; 100× transfection efficiency in vitro	[114,115,116]
Polymeric Nanoparticles	HER2, H-ferritin, CEA, ICAM-1, Staphylococcus aureus	Cancer (breast, colorectal), infectious diseases, GI inflammation	Paclitaxel, docetaxel, antibiotics	Enhanced tumor targeting and cytotoxicity; improved oral GI delivery and bacterial clearance	[127,128,142,143,144]
Protein Nanoparticles	VEGF, CD3, CD40	Gynecologic cancers, T-cell leukemia, and viral infections	Paclitaxel, p53, immune modulators	50% increased drug delivery; specific immune cell targeting; enhanced immune signaling via nanocages	[132,133,145]
Gold/Metal Oxide Nanoparticles	EGFR, HER2, EpCAM, VEGF, CD105, NNV, CD63	Cancer (breast, colorectal, liver), cardiac repair, viral infections	Paclitaxel, doxorubicin, bortezomib, mRNA	Photothermal killing; enhanced imaging; tumor targeting; 40× exosome redirection to heart	[136,137,138,139,140,146,147,148,149,150,151,152]

## Data Availability

No new data were created or analyzed in this study.

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
