# Peer review of "Strategic and Chemical Advances in Antibody–Drug Conjugates"

_pharmaceutics, 2025, doi:10.3390/pharmaceutics17091164_

Round 1

Reviewer 1 Report

Comments and Suggestions for Authors

The review focuses on antibody-drug conjugates, covering their historical development, molecular design and composition, describes their application in cancer and in other areas, as well as various delivery systems using ADCs. The manuscript is generally well written. I believe that addressing the following points would further improve its quality:

  1. Page 3, Introduction, sentence “Many of the prevalent and successful FDA-approved ADCs use either monomethyl auristatin E [MMAE, AKA (DM1)] or calicheamicin as their payload [14].” MMAE and DM1 belong to different structural classes. DM1 is a maytansine derivative, while MMAE is a dolastatin derivative. They also bind to different binding sites on tubulin (DM1 targets the maytansine site, MMAE targets the vinca alkaloid site). Therefore, MMAE and DM1 should be discussed separately, not under the same category.
  2. Page 9, sentence “By 2025, the ADC landscape will include 19 FDA-approved products and over 200 clinical candidates, targeting a diverse range of antigens across hematologic and solid tumors.” Should it be corrected to “includes” instead of “will include”, as it is currently 2025.
  3. References should be thoroughly checked throughout the text, e,g, Reference 60 in the following sentence is about enzyme-based bioconjugation, not about click chemistry (and is used in the previous paragraph about bioconjugation): Page 19, sentence “Click chemistry offers numerous advantages for ADC synthesis, including mild reaction conditions, high yields, compatibility with aqueous and organic solvents, stereoselectivity, and straightforward product isolation, making it highly suitable for bioconjugation applications [60].“
  4. The same applies for the next sentence, where reference 58 is used, however, the referenced paper does not mention click chemistry: page 19 sentence “As a result, click reactions have become a powerful and efficient tool for bioconjugation, the process of covalently linking chemically distinct molecules, due to their versatility, reliability, and compatibility with complex biological systems [58].”
  5. On page 27, in the paragraphs about T-Dxd, T-DM1, and Brentuximab Vedotin, I suggest to start the description with T-DM1 as it was historically developed and approved first and then describe T-Dxd and compare them. Brentuximab Vedotin is logically not fitting here as it does not target HER2 but CD-30. It would better to move it to another section.

Author Response

  1. Page 3, Introduction, sentence “Many of the prevalent and successful FDA-approved ADCs use either monomethyl auristatin E [MMAE, AKA (DM1)] or calicheamicin as their payload [14].” MMAE and DM1 belong to different structural classes. DM1 is a maytansine derivative, while MMAE is a dolastatin derivative. They also bind to different binding sites on tubulin (DM1 targets the maytansine site, MMAE targets the vinca alkaloid site). Therefore, MMAE and DM1 should be discussed separately, not under the same category.

Response: We appreciate this important clarification. The sentence has been revised into a new paragraph to accurately reflect the structural and mechanistic differences between DM1 and MMAE. It now reads:

DM1 and MMAE are two of the most commonly used cytotoxic payloads in FDA-approved antibody-drug conjugates (ADCs), both acting as microtubule inhibitors that induce mitotic arrest and apoptosis. Despite their similar functional outcomes, they differ in origin, structure, and binding sites: DM1, a maytansine derivative, binds the maytansine site on β-tubulin, while MMAE, a synthetic analog of dolastatin 10, targets the vinca alkaloid site. These agents also differ in conjugation strategies, DM1 is typically linked via non-cleavable linkers (e.g., in T-DM1), whereas MMAE is delivered through cleavable dipeptide linkers (e.g., in brentuximab vedotin). These design choices influence intracellular release, stability, bystander killing, and therapeutic index, making the selection of payload–linker combinations a critical component of ADC optimization.

  1. Page 9, sentence “By 2025, the ADC landscape will include 19 FDA-approved products and over 200 clinical candidates, targeting a diverse range of antigens across hematologic and solid tumors.” Should it be corrected to “includes” instead of “will include”, as it is currently 2025.

Response: Thank you for pointing this out. The sentence has been updated to reflect the current year (2025) by changing the verb tense to:

“As of 2025, the ADC landscape includes 19 FDA-approved products and over 200 clinical candidates…”

  1. References should be thoroughly checked throughout the text, e,g, Reference 60 in the following sentence is about enzyme-based bioconjugation, not about click chemistry (and is used in the previous paragraph about bioconjugation): Page 19, sentence “Click chemistry offers numerous advantages for ADC synthesis, including mild reaction conditions, high yields, compatibility with aqueous and organic solvents, stereoselectivity, and straightforward product isolation, making it highly suitable for bioconjugation applications [60].“

Response: We apologize for the referencing error. Reference 60 have been carefully reviewed and corrected. The sentence on page 19 now cites appropriate literature specifically discussing click chemistry in the context of ADCs.

  1. The same applies for the next sentence, where reference 58 is used, however, the referenced paper does not mention click chemistry: page 19 sentence “As a result, click reactions have become a powerful and efficient tool for bioconjugation, the process of covalently linking chemically distinct molecules, due to their versatility, reliability, and compatibility with complex biological systems [58].”

Response: We apologize for the reference error. We have also ensured that the sentence is supported by a reference explicitly addressing the role of click reactions in bioconjugation.

  1. On page 27, in the paragraphs about T-Dxd, T-DM1, and Brentuximab Vedotin, I suggest to start the description with T-DM1 as it was historically developed and approved first and then describe T-Dxd and compare them. Brentuximab Vedotin is logically not fitting here as it does not target HER2 but CD-30. It would better to move it to another section.

Response: We fully agree with this logical sequencing. The paragraph has been reorganized to begin with T-DM1, followed by T-Dxd with comparative discussion, and Brentuximab Vedotin has been moved to the CD30-targeting ADCs section to better align with its biological target and clinical application.

Reviewer 2 Report

Comments and Suggestions for Authors

This review provides a comprehensive and up-to-date overview of antibody–drug conjugates (ADCs), covering their historical development, chemical strategies for conjugation, clinical progress, and future perspectives. It is exceptionally well written, logically structured, and in my opinion, one of the best reviews I have evaluated over the past year.

That said, there are a few minor issues that should be addressed before acceptance:

  1. Table 1 - ICâ‚…â‚€ values are reported in mixed units (mass-based and molar). These should be unified for consistency and clarity.

  2. Figure 1 - The labels are pixelated and difficult to read, and the acknowledgement of BioRender.com is missing. Both issues should be corrected.

  3. Most important - The rationale behind Section 6 (Nanoparticle-based delivery) is unclear. The section describes antibody-functionalized nanoparticles for targeted delivery, which are not ADCs but rather a distinct class of non-protein systems. While these approaches may indeed compete with ADCs in the future, they are fundamentally different entities. Moreover, the opening statement “Nanoparticles offer several advantages when used in conjunction with ADCs” does not align with the content that follows, which focuses instead on nanoparticle*antibody systems. Given the vastness of the nanoparticle-based drug delivery field, it is unrealistic to cover it within this review. I would therefore strongly recommend either removing Section 6 entirely or reducing it to a brief (maximum one page) discussion that highlights nanoparticles as a potential alternative to ADCs, without attempting a detailed review.

With these revisions, the manuscript will be significantly strengthened and ready for publication.

Author Response

That said, there are a few minor issues that should be addressed before acceptance:

  1. Table 1 - ICâ‚…â‚€ values are reported in mixed units (mass-based and molar). These should be unified for consistency and clarity.

Response: Thank you for the observation. All ICâ‚…â‚€ values in Table 1 have been standardized to nanomolar (nM) units for consistency and clarity across the table. Where necessary, conversions from μg/mL to nM were performed using the molecular weights of the compounds.

  1. Figure 1 - The labels are pixelated and difficult to read, and the acknowledgement of BioRender.com is missing. Both issues should be corrected.

Response: We have replaced Figure 1 with a high-resolution version, ensuring all labels are sharp and legible. Additionally, the caption has been updated to include the required attribution:

“Figure created by BioRender.com.”

  1. Most important - The rationale behind Section 6 (Nanoparticle-based delivery) is unclear. The section describes antibody-functionalized nanoparticles for targeted delivery, which are not ADCs but rather a distinct class of non-protein systems. While these approaches may indeed compete with ADCs in the future, they are fundamentally different entities. Moreover, the opening statement “Nanoparticles offer several advantages when used in conjunction with ADCs” does not align with the content that follows, which focuses instead on nanoparticle*antibody systems. Given the vastness of the nanoparticle-based drug delivery field, it is unrealistic to cover it within this review. I would therefore strongly recommend either removing Section 6 entirely or reducing it to a brief (maximum one page) discussion that highlights nanoparticles as a potential alternative to ADCs, without attempting a detailed review.

Response: We thank the reviewer for this critical insight. In response, Section 6 has been significantly reduced to a concise summary (~1 page), now titled:

“Emerging Alternatives: Antibody-Conjugated Nanoparticle Systems.”

The revised section briefly highlights how antibody-conjugated nanocarriers represent a parallel strategy to ADCs rather than a direct extension. It no longer attempts to review nanoparticle delivery comprehensively and acknowledges their distinction from classical ADCs.

With these revisions, the manuscript will be significantly strengthened and ready for publication.